# Relative Individual Sprint in Most Demanding Passages of Play in Spanish Professional Soccer Matches

**DOI:** 10.3390/sports11040072

**Published:** 2023-03-23

**Authors:** Juan Ángel Piñero, Marcos Chena, Juan Carlos Zapardiel, Alberto Roso-Moliner, Elena Mainer-Pardos, Miguel Lampre, Demetrio Lozano

**Affiliations:** 1CF Fuenlabrada, 28942 Madrid, Spain; 2Facultad de Ciencias del Deporte de Toledo, Universidad de Castilla La Mancha, 13001 Ciudad Real, Spain; 3Faculty of Medicine and Health Sciences, Campus Universitario-C/19, University of Alcalá, Av. de Madrid, Km 33,600, 28871 Madrid, Spain; 4Faculty of Health Sciences, San Jorge University, Autovía A23 Km 299, 20830 Villanueva de Gállego, 50830 Zaragoza, Spain

**Keywords:** football, GPS, worst-case scenario, match analyse, sprint distance

## Abstract

(1) Background: The objective of this research was to analyse the most demanding passages (MDP) considering the sprint variable relative to the maximum level of sprint ability of each player as a function of player position, final outcome and part of the match during the competitive phase of a professional soccer season. (2) Methods: Global positioning system (GPS) data were collected from 22 players according to their playing position in the last 19 match days of the Spanish La Liga professional soccer in the 2020/2021 season. MDP were calculated from 80% of the maximum sprint speed of each player. (3) Results: Wide midfielders covered the greatest distance at >80% of the maximum speed (2.4 ± 1.63 seg) and the longest duration (21.91 ± 13.35 m) in their MDP. When the whole team was losing, it demonstrated greater distances (20.23 ± 13.04 m) and longer durations (2.24 ± 1.58 seg) compared to games in which it was winning. When the team ended up drawing, the relative sprint distance covered in the second half was significantly greater than in the first (16.12 ± 21.02; SD = 0.26 ± 0.28 (−0.03/−0.54). (4) Conclusions: Different demands of MDP, according to the sprint variable relative to the maximum individual capacity in competition, are required when contextual game factors are considered.

## 1. Introduction

In modern soccer, technological advances and match and training analyses have provided coaches with scientifically based data on player performance with respect to the physical, technical and tactical dimensions of the game [1,2]. Quantifying and knowing the physical demands during competition is of great importance to better understanding what happens during competition and allowing for the adequate management of training intensity [3,4] through the adequate periodization of specific tasks. An optimal dose of training helps to conditionally prepare the soccer player for competition while minimizing the incidence of overuse injuries [5,6]. In recent years, high-intensity actions and the distance and number of sprints have increased by approximately 30–35% in official competition matches [7]. Therefore, players need to be robust enough to cope with such demands [8] and execute their tactical role effectively [9].

Moreover, to augment the intensity of the game, the rate and burden of hamstring injuries has also increased significantly since 2001, showing an average annual increase of 2.3% [10]. Although its origin is multifactorial [11], the increase in hamstring injuries could be logical as an increase in competitive demands since 70% of these injuries usually occur during sprints or high-speed runs [12]. This evolution of the game towards high-intensity, repetitive actions increases the risk of hamstring injuries [10] due to the presence of high-force, repetitive eccentric actions [13]. Additionally, reduced recovery between these high-intensity efforts [14] may be associated with an increased risk of injury.

For this reason, control and knowledge of the most demanding passages (the most intense periods of training, or matches that show peak activities higher than the average of the match) associated with sprinting can be very useful for attempting to minimize the risk of injury and, in this way, trying to prevent hamstring injuries in soccer due to the negative impact that injuries have on soccer teams at an economic level and with respect to team performance [15,16,17]. Our research deals precisely with this issue, using knowledge of the most demanding passages related to individualized maximum sprint capacity to attempt to minimize or reduce the hamstring injury rate in soccer teams, allowing coaches and trainers to prescribe optimal training loads in order to prepare athletes to meet the needs of the competition.

Average physical demands, traditionally used as the main method of load quantification, could be underestimating the most demanding passages (MDP) to which players are subjected during the match [18,19,20,21] due to the intermittency of the game itself [22,23]. The MDP is the most intense period of training or matches and demonstrates peak activities that are higher than the average of the match [18,19]. The rolling average method has been one of the main instruments used to analyse the MDP [4,8,18,24,25] in different interval windows (1, 3, 5 and 10 min). The intensity of the MDP is greater when the duration of the studied period is shorter [19]

The unpredictable nature of soccer has demonstrated that the physical demands are influenced by tactical–technical factors [26]. MDP are contextually dependent, as research has shown that they can vary based on the player-specific position on the pitch [4,8,27,28,29], tactical role [27], play formation [30], match outcome [29], whether the team is playing at home or away [29], the first or second half of the game [4,29] and match congestion period [24].

However, despite the importance of top-speed actions in soccer, there is currently no consensus on the definition of sprinting. This is because there are different criteria regarding the defined speed thresholds in soccer [31,32,33]. The players’ MDP is over- or underestimated depending on their sprint ability. Concerning MDP calculated with relative thresholds, the MDP values are overestimated for faster players, and underestimated for slower players [31]. To date, no known manuscript reports information on the MDP of the competition based on the maximum capacity of soccer players, which one might think is information of relative interest from a practical perspective [34]. Regarding these issues, this study has the novelty of analysing the efforts of maximum demands relative to the maximum individual sprinting capacity of each soccer player. The objective of this study was to analyse the most demanding passages (MDP), considering the sprint variable relative to the maximum level of sprint ability of each player as a function of player position, final outcome and part of the match during the competitive phase of a professional soccer season.

## 2. Materials and Methods

The design responds to a retrospective observational research study carried out with professional soccer players during a season. The MDP of each player was analysed depending on his maximum speed in professional soccer matches. The independent variables used were the playing position, the halftime of the match, the result of the match and the competition phase. Global positioning system (GPS) data were collected from 22 players in a total of 19 games (90 min plus extra time) from the last 19 match days in the 2020–2021 competitive season. The players provided informed consent via signature, and the study was fully approved by the Sports Management Department of the Football Club. In addition, the study complied with the research ethics standards of the University of Alcalá, Madrid, Spain, code CEI/HU/2019/08, and was conducted by the principles set out in the Declaration of Helsinki.

### 2.1. Participants

Twenty-two professional soccer players participated in this study (age: 25.8 ± 5 years; mass: 76.4 ± 6.5 kg; stature: 1.81 ± 0.07 m). The players belonged to a club in the Spanish La Liga professional soccer at Tier 4: the Elite/International Level [35]. Data were collected throughout 19 competitive matches in the 2020–2021 competitive season (6 wins; 6 draws; 7 losses; final position: 11th). The players were grouped according to their playing positions as central defenders (CD: *n* = 4), fullbacks (FB: *n* = 4), midfielders (MF: *n* = 6), wide midfielders (WM: *n* = 4) and forwards (FW: *n* = 4). Goalkeepers were not included in the analysis.

### 2.2. Reseach Instruments

The time motion of each player was recorded individually in all matches with a 10 Hz GPS device, as used in previous studies [36,37]. Research has shown this system to be a valid and reliable assessment for monitoring the movement demands of team players [38]. To avoid variability, each player always used the same GPS device, a WIMU PRO (RealTrackSystem, Almería, Spain), located between the two scapulae with a special vest. The same methodology used in earlier studies in which the MDP of a professional soccer game was adopted for data collection [4,8,9,24]. The coding and analysis of the data were performed with sPRO software (RealTrackSystems, Almeria, Spain) [39]. To properly monitor and prescribe high-intensity training loads, predefined (absolute) or individualized (relative) criteria were used [32,33]. Although most of the published information determines the performance profile in soccer through absolute thresholds, it has been shown that these thresholds could overestimate or underestimate the amount of high- and maximum-intensity efforts of the players [32,33]. Thus, MDP were calculated from 80% of the maximum sprint speed of each player. Moreover, monitoring the distance covered for >80% of the maximum speed guarantees greater precision in the determination of sprint workloads and the associated injury risk [32]. Analysis was performed using time windows of 1 min because speeds above 80% of the maximum speed of each player are hardly bearable for longer periods [40]. This study analysed MDP during short-duration game actions. For this purpose, the range of scenarios recorded for each match was set at 50% of the maximum obtained for the selected variable, thus filtering out the cases of maximum demand scenarios in each match.

### 2.3. Procedures

All players performed two control tests at three months apart (September–January): the 40 m linear maximal speed test [41,42,43]. Maximum speed data were also recorded at every training session and match during the study (January–May) to monitor the variation in each player’s maximum speed throughout the season. Each match was 90 min in duration (two 45 min halves), plus extra time (4–9 min). Data were collected based on what happened in each of the halves, the positions of the players and the result. These data were then averaged across all observations by position for the analysis between groups concerning the distance travelled and the duration of each MDP.

### 2.4. Data Analysis

All data are presented as means and standard deviations (mean ±SD). Magnitude-based inferences and a precision of estimation were used to analyse the data [44]. All processed variables were log-transformed to reduce the non-uniformity of error. Differences between the halves (first half and second half) were assessed via standardised mean differences (Cohen’s d) and respective 90% confidence limits. The effect size was calculated using the Cohen’s d from the differences of the groups’ means and the weighted standard deviation. Threshold values for standardized differences were >0.2 (small), >0.6 (moderate), >1.2 (large) and very large (>2.0) [45]. The statistical analysis was performed with the software package SPSS, version 28.0 (SPSS Inc., Chicago, IL, USA).

## 3. Results

Table 1 shows the mean ± SD and confidence intervals (95%) duration (s), distance (m·min^−1^) and maximum speed (m·s^−1^) of all the MDP values for each position of the players.

Table 2 shows the mean ± SD and confidence intervals (95%), duration (s), distance (m·min^−1^) and maximum running velocity (m·s^−1^) of all the MDP values for each half of the match.

Table 3 shows the mean ± SD and confidence intervals (95%), duration (s), distance (m·min^−1^) and maximum speed (m·s^−1^) of all the MDP values according to the result of the match.

Table 4 shows the mean ± SD and confidence intervals (95%), duration (s), distance (m·min^−1^) and maximum speed (m·s^−1^) of all the MDP values according to the date of the season.

Table 5 shows the difference standardized (Cohen) of the MDP according to position on the half of the match.

Table 6 shows the difference standardized (Cohen) of the MDP according to the result of the match on the half of the match.

Figure 1 shows effect size on the MDP according to the position and half of the match.

Figure 2 shows the effect size on the MDP according to the result of the match.

## 4. Discussion

The objectives of the present study were: (i) to analyse the MDP, considering the sprint variable relative to the maximum level of sprint ability of each player during the competitive phase of a professional soccer season; (ii) to compare the MDP of each playing position; and (iii) to compare the MDP between the first and second halves of each match. This is the first study to analyse the efforts of maximum demands relative to the maximum individual capacities of each soccer player.

Competitive demands have led coaches and physical trainers to challenge the principles of training through the prescription of optimal stimuli to develop or maintain adaptations without exceeding the limits of physiological tolerance of the players [36,46]. For this purpose, the MDP must be considered in the training context since planning according to the average competition references may not be a sufficient stimulus to prepare soccer players for the most demanding phases [8,47]. To describe the MDP, the calculation of the moving average has been considered the most accurate method for determining the most intense periods [8,34], as it was calculated in this study. However, most authors have focused their study on describing the MDP in terms of the variables total distance per minute, distance covered at a high intensity (>19.8 km/h) per minute and distance covered at a sprint (>25.2 km/h) per minute in periods of 1, 3, 5 and 10 min [8,27,28,34].

Unlike the works mentioned above, this study attempted to analyse the sprint MDP by considering sprint the ability to make a maximum effort or very close to the maximum individual capacity of each player to obtain the distance, duration time and speed information in those competitive situations. Of the locomotor requirements demanded by the game, the sprint is probably one of the most determining since it precedes most scoring opportunities and is the most frequent mechanism of muscle injury.

Players have been shown to reach 85–95% of their maximum speed during soccer matches [46]. Therefore, it appears that training conditions of maximal stress based on an individual soccer player’s ability to optimally meet competitive demands is required [46,48,49]. On the other hand, O’Connor et al. (2020) [32] demonstrated significant differences in the results obtained when they analysed the relationship between the sprint workload, analysed with relative and absolute thresholds, and the incidence of injuries, concluding that monitoring the distance covered at >80% of the maximum speed guarantees greater precision in the determination of sprint workloads and associated injury risk. Accordingly, and considering that the fitness of athletes is variable throughout the competitive period, in this study, it was decided to control the sprint variable using the relative threshold of >80% of the maximum individual speed reached in competition, which was automatically updated to the maximum value obtained throughout the season. However, in some cases, such as for players with a certain time of inactivity or low participation and poor competitive rhythm, this value of maximum speed obtained at a specific moment would be higher than the real fitness that these players have for the speed. For this reason, a relative range of 80% of the maximum sprint was placed for those players whose top speed was overestimated due to their low fitness to determinate a more accurate MDP value relative to the maximum speed at this particular moment, guaranteeing greater precision in the determination of sprint workloads and the associated injury risk.

This study showed that during the MDP in the matches, there were differences between the different positions, as observed in previous studies [8,18,29]. The results showed that wide midfielders were the players who covered the greatest distance at >80% of the maximum speed in their MDP and demonstrated the longest duration of these efforts. According to the scientific literature, wide midfielders undergo greater distances at a high intensity (HSR) during competition [26,50,51,52,53,54]. Nonetheless, Martín-García and Gómez Díaz et al. (2018) [8] did not find the highest HSR values in the MDP of these players. These authors warned that their results could have been different if they had used the HSR as a criterion variable, as it was used in other publications [18]. However, the maximum capacities of the athletes were not taken into account in any of the studies that employed HSR as a criterion [33], registering said variable with absolute criteria.

Although midfielders must cover great distances throughout a game [26,51,52,54], when the total distance was analysed regardless of speed, these players were the ones who covered the most meters per minute [8]. In this study, it was observed that according to their MDP, the distance they covered in the relative sprint was less than other positions. Therefore, these results confirm that midfielders are players with a high level of intervention in game actions and are capable of recovering great distances but do not have the need to reach the maximum demands of their capacity [51].

In the analysis of the defensive line, there were differences in the MDP regarding the distance covered in a relative sprint between halves. Fullbacks covered more in the first part than the second; however, central defenders covered a significantly greater distance and duration relative to the sprint in the second half. In line with these results, Rey et al. (2020) reported that central defenders increased the sprint distance covered during the second half of the match [55]. Soccer matches tend to present less control and higher transition situations at the end of the matches due to the need to win. This could explain the fact that central defenders covered greater distances and durations relative to their sprint, since central defenders demonstrate most of their high intensity actions in recovery runs and coverage [56], defensive situations that occur constantly in transition situations. In our study, no statistically significant differences were found between both parties in the rest of the positions (wide midfielder and forward).

Moreover, analysing the competitive physical demands of soccer players based on the total playing time could overestimate fatigue-induced decreases in performance. Rey et al. (2020) [55] found that there were no significant differences when the whole team is taken as a reference, without differentiating between positions, in the efforts made by the professional soccer players between the first and the second parts, taking into account the useful time of exposure. However, it seems that the physical demands could be positionally dependent, since when we differentiate between positions, central defenders and central midfielders increased the sprint distance covered during the second half of the match [55]. Therefore, the effective playing time and playing position should be taken into account when analysing the physical demands and performance of soccer players.

Regarding the relationship with the final score, Oliva-Lozano et al. (2020) [29] exposed that the result was a contextual variable with a significant impact on the most demanding competitive scenarios in total distance, high-speed running and sprint distance in the periods of 1 and 3 min. Winning the match resulted in higher values of total distance covered, the distance at high intensity and the distance at sprint compared to drawing and losing the game [29]. Diez et al. (2021) [57] also demonstrated that more SPR are exhibited by players when the team plays at home and wins. However, in contrast to these studies, our results indicated that games in which the team was losing showed greater distances and longer durations in maximum efforts compared to games in which it was winning. The timing of the matches selected in this study included the last match weeks of the season (J19 to J42), in which the teams fight for their objectives and each point won or lost has greater relevance. Perhaps this would justify the results of our study, since when teams are losing, they need to make greater physical efforts to reduce that difference and try to win. Moreover, in line with these results, when considering the MDP of the matches in which the team ended up drawing, the relative sprint distance covered in the second half was significantly greater than in the first, perhaps due to the need to score to achieve the classifying objectives. Nobari et al. (2021) [58] also highlighted the difference between drawn versus won matches, explaining that the similar performance may be related to the duration of the match and therefore the intensity generated in both.

For the whole team, a greater duration and distance for the MDP of the sprint covered in the second part could have an impact on hamstring injuries since the hamstring fatigue resistance has been shown to be a soccer-specific risk factor for this type of injury [59,60]. In line, Raya et al. (2020) [61] reported that the 76–90 min time period presented the highest values of hamstring injury: 12 out of 63 injuries (19.04% of total) in comparison to all selected time periods in an Elite Spanish Male Academy. Sprinting is a key area for improving performance and injury prevention. The authors of [46,50,62] showed that professional soccer players who were trained to a higher chronic training load were able to better tolerate a weekly acute increase in exposure to maximum speed events as a consequence of the protective effect of chronic, high training loads. Therefore, these data suggest the need for athletes to regularly expose themselves during training to maximum speeds [32,46] and in fatigue conditions to prepare for competition. This could help coaches plan and structure their training and competitions and try to minimize or reduce the injury rate.

Future research on the relationship between sprinting at relative thresholds and MDP linked to training tasks that can replicate competitive demands seems necessary, with the aim of optimizing the training process to improve performance and prevent injuries.

## 5. Conclusions

The physical performance profile in soccer should not be understood solely from the average analysis of competitive demands, since this could be hiding the maximum demands covered by players. Players do not show a decrease in physical performance as the match progresses, even increasing their performance in situations of maximum demand in the second half. Moreover, other contextual factors as the result of the match, playing position and the moment of the season can influence the MDP.

Coaches and trainers must prescribe optimal sprint training loads, prioritizing both quality and quantity in situations of fatigue, due to the increased duration and distance in the MDP of the sprint covered in the second half, as demonstrated by this study, in order to prepare athletes for meeting competitive needs and to reduce the risk of hamstring injury. Moreover, these tasks should be oriented to individual and positional needs, due to the tactical and positional dependence on physical demands.

It is therefore necessary to detect which are the periods of maximum demand during the matches and which of them can be generated through specific training tasks to replicate the demands of the competition by taking the athlete to situations at the limit of his capacity.

The main practical application of knowing these demands is the ability to apply and practice training tasks that meet these needs, both individually and positionally, on a day-to-day basis, not only to improve performance but also to have a protective effect on the relationship that sprinting has been demonstrated to have with both.

## Figures and Tables

**Figure 1 sports-11-00072-f001:**
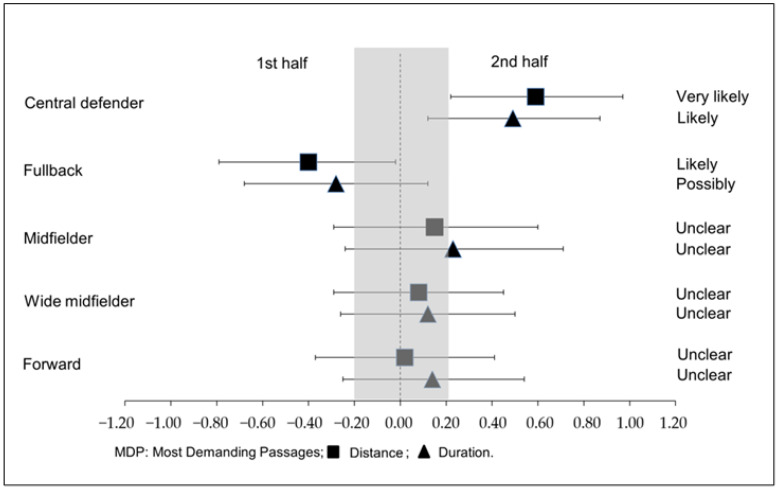
Effect size to the Most Demanding Passages according to position and half of the match.

**Figure 2 sports-11-00072-f002:**
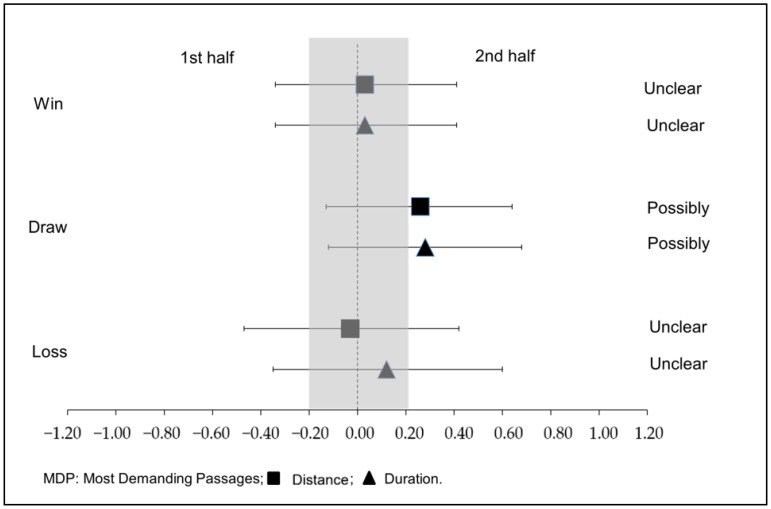
Effect size on the most demanding passages according to the result of the mach.

**Table 1 sports-11-00072-t001:** Duration, distance, and maximal running velocity to the most demanding passages according to play position.

Part of the Match		MDP								
		Duration (s)			Distance (m)			Vmax (m/s)		
	*n*	Values ± SD	IC 95%		Values ± SD	IC 95%		Values ± SD	IC 95%	
**1º Part**	151	2.05 ± 1.37	1.83	2.27	18.98 ± 11.42	17.15	20.82	32.15 ± 1.63	31.89	32.41
**2º Part**	192	2.28 ± 1.64	2.05	2.52	20.57 ± 13.67	18.62	22.51	32.17 ± 1.61	31.94	32.40

MDP: Most Demanding Passages; Vmax: Maxima velocity in competition.

**Table 2 sports-11-00072-t002:** Duration, distance, and maximal running velocity of the most demanding passages according to part of the match.

Position		MDP								
		Duration (s)			Distance (m)			Vmax (m/s)		
	*n*	Values ± SD	IC 95%		Values ± SD	IC 95%		Values ± SD	IC 95%	
**Central defender**	75	2.15 ± 1.48	1.81	2.49	19.26 ± 12.01	16.47	22.04	31.74 ± 1.79	31.33	32.15
**Fullback**	69	2.08 ± 1.37	1.75	2.41	19.31 ± 11.87	16.46	22.16	32.30 ± 1.11	32.04	32.57
**Midfielder**	48	2.13 ± 1.53	1.68	2.57	18.75 ± 12.29	15.22	22.36	30.55 ± 1.41	30.14	30.96
**Wide midfielder**	82	2.4 ± 1.63	2.04	2.76	21.91 ± 13.35	18.98	24.85	32.68 ± 1.74	32.29	33.06
**Forward**	69	2.10 ± 1.61	1.71	2.49	19.42 ± 13.87	16.09	22.75	32.99 ± 0.68	32.82	33.15

MDP: Most Demanding Passages; Vmax: Maxima velocity in competition.

**Table 3 sports-11-00072-t003:** Duration, distance, and maximal running velocity for the most demanding passages according to result of the match.

Result		MDP								
		Duration (s)			Distance (m)			Vmax (m/s)		
	*n*	Values ± SD	IC 95%		Values ± SD	IC 95%		Values ± SD	IC 95%	
**Win**	115	2.05 ± 1.40	1.79	2.31	18.77 ± 11.76	16.60	20.94	32.11 ± 1.62	31.81	32.41
**Draw**	128	2.25 ± 1.59	1.98	2.53	20.57 ± 13.36	18.24	22.91	32.30 ± 1.62	32.02	32.59
**Loss**	100	2.24 ± 1.58	1.92	2.55	20.23 ± 13.04	17.65	22.82	32.03 ± 1.61	31.71	32.35

MDP: Most Demanding Passages; Vmax: Maxima velocity in competition.

**Table 4 sports-11-00072-t004:** Duration, distance, and maximal running velocity to the most demanding passages according to date of the season.

Date		MDP								
		Duration (s)			Distance (m)			Vmax (m/s)		
	*n*	Values ± SD	IC 95%		Values ± SD	IC 95%		Values ± SD	IC 95%	
**J24**	18	1.97 ± 1.61	1.17	2.77	18.21 ± 13.58	11.45	24.96	32.11 ± 1.70	31.27	32.95
**J25**	21	2.48 ± 1.28	1.87	3.08	22.56 ± 11.39	17.23	27.90	32.51 ± 1.73	31.70	33.32
**J26**	17	1.39 ± 1.01	0.85	1.93	12.73 ± 8.96	7.96	17.51	32.42 ± 1.84	31.44	33.40
**J27**	18	2.66 ± 1.47	1.91	3.42	23.99 ± 11.91	23.99	17.87	32.37 ± 1.75	31.37	33.27
**J28**	21	2.30 ± 1.32	1.70	2.90	21.62 ± 10.70	16.75	26.50	32.29 ± 1.86	31.44	33.13
**J29**	19	1.97 ± 1.18	1.97	2.55	17.99 ± 8.98	13.52	22.45	31.93 ± 1.56	31.15	32.71
**J30**	15	1.46 ± 1.12	0.81	2.11	14.09 ± 8.95	8.93	19.26	31.85 ± 1.29	31.11	31.58
**J31**	22	2.14 ± 1.59	2.14	1.41	20.14 ± 12.99	14.22	26.06	31.92 ± 1.49	31.24	32.60
**J32**	13	2.02 ± 1.60	1.00	3.03	18.81 ± 12.26	11.02	26.60	31.80 ± 1.79	30.67	32.94
**J33**	17	1.81 ± 1.31	1.11	2.51	16.41 ± 10.40	10.86	21.95	31.91 ± 1.34	31.20	32.63
**J34**	21	2.34 ± 2.10	1.39	3.30	20.58 ± 15.99	13.30	27.86	32.31 ± 1.37	31.69	32.94
**J35**	18	2.50 ± 1.68	1.64	3.36	22.93 ± 14.53	15.45	30.40	32.49 ± 1.34	31.80	33.18
**J36**	21	2.06 ± 1.38	1.41	2.71	16.74 ± 10.69	11.73	21.74	32.26 ± 1.69	31.47	33.05
**J37**	21	2.09 ± 1.58	1.33	2.86	19.95 ± 13.57	13.41	26.49	32.24 ± 1.59	31.47	33.00
**J38**	19	1.79 ± 1.25	1.19	2.40	16.19 ± 9.54	11.59	20.79	32.17 ± 1.54	31.43	32.91
**J39**	17	3.58 ± 1.92	2.56	4.61	31.95 ± 16.87	22.96	40.94	31.91 ± 1.76	30.97	32.85
**J40**	20	2.30 ± 1.66	1.50	3.10	20.66 ± 13.76	14.03	27.29	32.34 ± 1.84	31.46	33.23
**J41**	19	2.54 ± 1.56	1.76	3.32	23.22 ± 14.83	15.84	30.59	32.18 ± 1.67	31.35	33.01
**J42**	22	1.90 ± 1.33	1.30	2.51	17.78 ± 11.47	12.55	23.00	31.85 ± 1.88	31.00	32.71

MDP: Most Demanding Passages; Vmax: Maxima velocity in competition.

**Table 5 sports-11-00072-t005:** Difference standardized (Cohen) to the Most Demanding Passages according half of the mach.

Position	MDP	1st Half	2nd Half	Difference (%)	Difference Standardized (Cohen)	Chances	Qualitative
**Central defender**	Distance (m)	15.36 ± 10.54 (36)	22.30 ± 12.77 (39)	31.14 ± 17.46	0.59 ± 0.38 (0.21/0.96)	95/5/0	Very Likely
	Duration (s)	1.77 ± 1.26 (36)	2.48 ± 1.59 (39)	28.43 ± 20.52	0.49 ± 0.38 (0.11/0.86)	89/10/0	Likely
**Fullback**	Distance (m)	21.17 ± 11.03 (39)	16.40 ± 1.41 (33)	29.12 ± 13.24	−0.40 ± 0.39 (−0.78/-0.01)	1/19/80	Likely
	Duration (s)	2.28 ± 1.33 (39)	2.19 ± 1.83 (33)	20.62 ± 5.28	−0.28 ± 0.40 (−0.68/0.12)	2/34/63	Possibly
**Midfielder**	Distance (m)	15.53 ± 10.47 (26)	17.44 ± 14.90 (22)	10.93 ± 29.72	0.15 ± 0.44 (−0.30/0.59)	42/48/10	Unclear
	Duration (s)	1.94 ± 1.16 (26)	2.29 ± 1.81 (22)	15.21 ± 35.95	0.23 ± 0.47 (−0.25/0.70)	54/40/7	Unclear
**Wide midfielder**	Distance (m)	20.00 ± 14.23 (32)	21.18 ± 13.67 (51)	5.56 ± 4.14	0.08 ± 0.37 (−0.29/0.45)	30/60/10	Unclear
	Duration (s)	2.28 ± 1.70 (32)	2.47 ± 1.60 (51)	7.94 ± 6.42	0.12 ± 0.38 (−0.26/0.50)	36/56/8	Unclear
**Forward**	Distance (m)	19.25 ± 11.33 (30)	19.55 ± 15.69 (39)	1.51 ± 27.77	0.02 ± 0.39 (−0.37/0.41)	22/60/18	Unclear
	Duration (s)	1.98 ± 1.31 (30)	2.19 ± 1.83 (39)	9.94 ± 28.68	0.14 ± 0.39 (−0.26/0.53)	39/53/8	Unclear

MDP: Most Demanding Passages.

**Table 6 sports-11-00072-t006:** Difference standardized (Cohen) of the most demanding passages according to result of the match and the half of the match.

Results	MDP	1st Half	2nd Half	Difference (%)	Difference Standardized (Cohen)	Chances	Qualitative
**Win**	Distance (m)	17.56 ± 11.43 (54)	17.93 ± 12.76 (66)	2.11 ± 10.44	0.03 ± 0.30 (−0.27/0.33)	17/72/10	Unclear
	Duration (s)	2.03 ± 1.29 (54)	2.07 ± 1.50 (66)	2.24 ± 14.03	0.03 ± 0.31 (0.27/0.34)	18/71/11	Unclear
**Draw**	Distance (m)	18.02 ± 11.65 (56)	21.48 ± 14.76 (74)	16.12 ± 21.02	0.26 ± 0.28 (−0.03/-0.54)	63/36/0	Possibly
	Duration (s)	2.01 ± 1.38 (39)	14.76 ± 1.72 (74)	17.89 ± 19.61	0.28 ± 0.29 (−0.01/0.57)	67/32/0	Possibly
**Loss**	Distance (m)	19.56 ± 12.31 (45)	19.19 ± 14.14 (59)	1.93 ± 12.99	−0.03 ± 0.31 (−0.35/0.29)	12/69/19	Unclear
	Duration (s)	2.14 ± 1.46 (45)	2.32 ± 1.69 (59)	8.12 ± 13.52	0.12 ± 0.33 (−0.21/0.45)	34/60/6	Unclear

MDP: Most Demanding Passages.

## Data Availability

Not applicable.

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
