# Peer review of "Relative Individual Sprint in Most Demanding Passages of Play in Spanish Professional Soccer Matches"

_sports, 2023, doi:10.3390/sports11040072_

Round 1
Reviewer 1 Report
Dear Authors,
It has been a pleasure to read you work aimed at the analysis of the most demanding passages of each player during the competitive phase of a professional soccer season.
I believe the work poses an interesting point of view on athlete’s health promotion, since the increase in hamstring injuries could be due to the increased competitive demands in the last years. Indeed, most of these injuries seems to occur during sprints or high-speed runs.
However, some further methodological details are needed, and some paragraphs are hard to follow and should be rephrased. Lastly, a thorough English language revision is suggested.
My point-by-point comments and suggestions are also uploaded in the comments to the author’s section.
Abstract, line 16: it is indicated that you analysed the last 19 matchdays, but there is no indication of which competitive season has been considered. Please add this information.
Abstract, Lines 19-24: at first you report results for the wide midfielders, but in the following paragraph you report the results starting with “When the team was losing showed…” Do you still refer to results of the wide midfielders or of the whole team? Please specify.
Lines 52-54: “Due to the great impact that injuries have in soccer, both economically and at the level of team and soccer player performance [16–18].” This sentence is hard to follow, please rephrase.
Lines 74-76: “It is shown that faster players overestimate the MDP values, while slower players underestimate the MDP values concerning the MDPs calculated with the relative thresholds [32].” It seems that you mean that the players estimate their own MDP. Maybe you wanted to say that the MDP of the players is over or under estimated depending on their sprint ability. Please clarify.
Line 77: it should be specified that the maximum individual capacities is the capacity of sprinting.
Line 79: same as before, it should be specified that the maximum level of each player is its sprint ability.
Lines 96-97: I guess that the Spanish La Liga professional soccer is the highest soccer level in the country, but it should be specified. Moreover, a more definition of the athlete’s competitive level could be provided following the indication of this work: McKay AKA, Stellingwerff T, Smith ES, Martin DT, Mujika I, Goosey-Tolfrey VL, Sheppard J, Burke LM. Defining Training and Performance Caliber: A Participant Classification Framework. Int J Sports Physiol Perform. 2022 Feb 1;17(2):317-331. doi: 10.1123/ijspp.2021-0451.
Lines 108-109: “For data collection, it was established based on the same procedure as in previous research on the MDP of the game in professional soccer [5,9,10,25].” This sentence is hard to follow, please rephrase.
Lines 110-111: “MDPs were calculated from 80% of the maximum sprint speed of each player.” You should explain the rationale for choosing this intensity, please provide the reference to justify the choice of the relative intensity.
Line 119: as before, here too you should explain the rationale for choosing the 40-meter linear maximal speed test. Please provide the reference to justify the choice of the test to asses maximum sprint ability in soccer players.
Lines 125-126: “The traditional model ignores the accelerated race [39].” I am sorry, but I really cannot understand what you mean with this statement, please provide some further information.
Lines 173- 179 and 184-188: the aim of the study is reported twice, please merge the two paragraph to avoid repetitions.
Lines 205-208: “Therefore, sprinting is a key area to improving performance and injury prevention [42,48,50] showed that professional soccer players with a higher chronic training load were able to tolerate increased exposures to maximum speed events as a consequence of a protective effect.” Here again, I cannot understand what you mean with this statement, please rewrite it providing some further information.
Line 224: maybe instead of the first “absolute” you meant “relative”.
Lines 225-227: here you explain the rationale for choosing 80% as the cutting point of the intensity, please report it also in the methods section.
Line 245: “meters per minute in periods of 1 minute”, it seems redundant to specify the period of 1 minute if the measure is meters per minute, maybe some further explanation could assist the reader.
Lines 250-253: this is really hard to follow, please rephrase it and shorten the periods.
Lines 254-263: this section of the discussion is quite confusing. It would be helpful if you ordered the comments on different positions and on the entire team separately.
Lines 264-265: Again, I am not sure what you mean by this statement. I would appreciate it if you rewrote it with more details.
Line 268-270: “However, in this study, it was shown that games in which the team was losing showed greater distances and longer durations in maximum efforts compared to games in which it was winning.” In my opinion, this result deserves some explanation and references to previous study.
Lines 277-286: the whole paragraph does not seem to be pertinent to your work. I suggest you either delate it or explain its relation with your results. In the latter case it should be completely rewritten because it is extremely difficult to understand.
Lines 287-293: it is difficult to follow the whole paragraph, it should be rewritten.
Lines 294-297: here again it is difficult to follow the paragraph, please rephrase it.
Lines 304-306: as before it is difficult to follow the paragraph, please rephrase it.
Lines 307-310: this seems to fit more in a “Practical application” section than in the “Conclusion”, you could offer some more example of the use of your results and provide a separate paragraph of practical applications.
Minor
Line 100: Wide midfielders, wide should not begin with a capital letter.
Line 264: I guess the full point after [25,30]. is a mistake. Otherwise, the sentence would make no sense.
Line 288: I imagine that “updated if both in training”, should be “updated it both in training”.
Along the text and in the tables, please be consistent with the unit of measurement style. If you use the (m.min-1) than also m/s should be m.s-1. If the latter is chosen, please write it in the proper form: m·min-1 and m·s-1.
Author Response
Dear Authors,
It has been a pleasure to read you work aimed at the analysis of the most demanding passages of each player during the competitive phase of a professional soccer season.
I believe the work poses an interesting point of view on athlete’s health promotion, since the increase in hamstring injuries could be due to the increased competitive demands in the last years. Indeed, most of these injuries seems to occur during sprints or high-speed runs.
However, some further methodological details are needed, and some paragraphs are hard to follow and should be rephrased. Lastly, a thorough English language revision is suggested.
My point-by-point comments and suggestions are also uploaded in the comments to the author’s section.
ABSTRACT
Abstract, line 16: it is indicated that you analysed the last 19 matchdays, but there is no indication of which competitive season has been considered. Please add this information.
Authors: Thank you very much for your contribution. Information added to the manuscript. Season 2020/2021.
Abstract, Lines 19-24: at first you report results for the wide midfielders, but in the following paragraph you report the results starting with “When the team was losing showed…” Do you still refer to results of the wide midfielders or of the whole team? Please specify.
Authors: Thank you very much for your contribution. Information specifies to the manuscript. The results refer to whole team.
INTRODUCTION
Lines 52-54: “Due to the great impact that injuries have in soccer, both economically and at the level of team and soccer player performance [16–18].” This sentence is hard to follow, please rephrase.
Authors: Thank you very much for your contribution. Paragraph rephrased: Due to the negative impact that injuries have in soccer teams at an economic level and on team performance [16–18].
Lines 74-76: “It is shown that faster players overestimate the MDP values, while slower players underestimate the MDP values concerning the MDPs calculated with the relative thresholds [32].” It seems that you mean that the players estimate their own MDP. Maybe you wanted to say that the MDP of the players is over or under estimated depending on their sprint ability. Please clarify.
Authors: Thank you very much for your contribution. Paragraph clarified. MDP of the players is over or under estimated depending on their sprint ability
Line 77: it should be specified that the maximum individual capacities is the capacity of sprinting.
Authors: Thank you very much for your contribution. Information added to the manuscript.
Line 79: same as before, it should be specified that the maximum level of each player is its sprint ability.
Authors: Thank you very much for your contribution. Information added to the manuscript
MATERIALS AND METHODS
Lines 96-97: I guess that the Spanish La Liga professional soccer is the highest soccer level in the country, but it should be specified. Moreover, a more definition of the athlete’s competitive level could be provided following the indication of this work: McKay AKA, Stellingwerff T, Smith ES, Martin DT, Mujika I, Goosey-Tolfrey VL, Sheppard J, Burke LM. Defining Training and Performance Caliber: A Participant Classification Framework. Int J Sports Physiol Perform. 2022 Feb 1;17(2):317-331. doi: 10.1123/ijspp.2021-0451.
Authors: Thank you very much for your contribution. Added in the manuscript. Players belonged to a club in the Spanish La Liga professional soccer, tier 4: elite/international Level (McKay et al., 2022).
Lines 108-109: “For data collection, it was established based on the same procedure as in previous research on the MDP of the game in professional soccer [5,9,10,25].” This sentence is hard to follow, please rephrase.
Authors: Thank you for catching the error. Paragraph rephrased: The same methodology used in earlier studies on the MDP of the game in professional soccer was adopted for data collecting.
Lines 110-111: “MDPs were calculated from 80% of the maximum sprint speed of each player.” You should explain the rationale for choosing this intensity, please provide the reference to justify the choice of the relative intensity.
Authors: Thank you very much for your contribution. Referenced added to the manuscript.
Line 119: as before, here too you should explain the rationale for choosing the 40-meter linear maximal speed test. Please provide the reference to justify the choice of the test to asses maximum sprint ability in soccer players.
Authors: Thank you very much for your contribution. Referenced added to the manuscript.
Lines 125-126: “The traditional model ignores the accelerated race [39].” I am sorry, but I really cannot understand what you mean with this statement, please provide some further information.
Authors: Thank you for catching the error. Irrelevant information. Removed from the manuscript
DISCUSSION
Lines 173- 179 and 184-188: the aim of the study is reported twice, please merge the two paragraph to avoid repetitions.
Authors: Thank you for catching the error. Irrelevant information. Removed from the manuscript to avoid repetitions
Lines 205-208: “Therefore, sprinting is a key area to improving performance and injury prevention [42,48,50] showed that professional soccer players with a higher chronic training load were able to tolerate increased exposures to maximum speed events as a consequence of a protective effect.” Here again, I cannot understand what you mean with this statement, please rewrite it providing some further information.
Authors: Thank you very much for your contribution. Paragraph rephrased: Therefore, sprinting is a key area to improving performance and injury prevention [46,52,54] showed that professional soccer players who were trained to a higher chronic training load were able to better tolerate weekly acute increased of exposures to maximum speed events as a consequence of a protective effect of chronic high loads of training.
Line 224: maybe instead of the first “absolute” you meant “relative”.
Authors: Thank you for catching the error. Removed absolute for relative.
Lines 225-227: here you explain the rationale for choosing 80% as the cutting point of the intensity, please report it also in the methods section.
Authors: Thank you very much for your contribution. Added to methods section
Line 245: “meters per minute in periods of 1 minute”, it seems redundant to specify the period of 1 minute if the measure is meters per minute, maybe some further explanation could assist the reader.
Authors: Thank you very much for your contribution. Eliminated redundant information
Lines 250-253: this is really hard to follow, please rephrase it and shorten the periods.
Authors: Thank you very much for your contribution. Paragraph rephrased: In the analysis of defensive line, there are differences in MDP regarding the distance covered in a relative sprint between halves. Fullbacks covered more in the first part concerning the second, however, central defenders covered the significantly greater distance and duration to their relative sprint in the second part. In line with this results, Rey et al. (2020), reported that central defenders, increased the sprint distance covered during the second half of the match [58]. Our study, in the rest of the positions (wide midfielder and forward) no statistically significant differences were found between both parties.
Lines 254-263: this section of the discussion is quite confusing. It would be helpful if you ordered the comments on different positions and on the entire team separately.
Authors: Thank you very much for your contribution. Paragraph rephrased: Moreover, analyzing the competitive physical demands of soccer players based on total playing time could overestimate fatigue-induced decreases in performance. Rey et al. (2020) found that there were no significant differences when whole team is taken as a reference, without differentiating between positions, in the efforts made by the professional soccer players between the first and the second part, taking into account the useful time of exposure. However, it seems that the physical demands could be positionally dependent, since, when we differentiate between positions, central defenders and central midfielders increased the sprint distance covered during the second half of the match [58], hence, effective playing time and playing position should be taken into account when analyze the physical demands and performance of soccer players
Lines 264-265: Again, I am not sure what you mean by this statement. I would appreciate it if you rewrote it with more details.
Authors: Thank you very much for your contribution. Rewrote more details: As for the relationship with the final score, Oliva-Lozano et al. (2020) exposed that the result was a contextual variable with a significant impact on the most demanding competitive scenarios in total distance, high speed running and sprint distance in the periods of 1 and 3 minutes
Line 268-270: “However, in this study, it was shown that games in which the team was losing showed greater distances and longer durations in maximum efforts compared to games in which it was winning.” In my opinion, this result deserves some explanation and references to previous study.
Authors: Thank you very much for your contribution. Paragraph modified
Lines 277-286: the whole paragraph does not seem to be pertinent to your work. I suggest you either delate it or explain its relation with your results. In the latter case it should be completely rewritten because it is extremely difficult to understand.
Authors: Thank you very much for your contribution. Paragraph deleted
Lines 287-293: it is difficult to follow the whole paragraph, it should be rewritten.
Authors: Thank you very much for your contribution. Paragraph rewritten: The present study takes as reference the maximum speed, obtained through a linear test and updated it both in training and in the competition, if this value is exceeded. However, in some cases as players with a certain time of inactivity or low participation and poor competitive rhythm, this value of maximum speed obtained at a specific moment, would be higher than the real fitness that these players have for the speed. Because of, can place a relative range of 80% maximum sprint for those players that top speed are overestimate due to his low fitness, to determinate more accurate his MDPs related to maximum speed in this particular moment and guarantees greater precision in the determination of sprint workloads and associated injury risk.
Lines 294-297: here again it is difficult to follow the paragraph, please rephrase it.
Authors: Thank you very much for your contribution. Paragraph rewritten: Future research on the relationship between sprinting in relative thresholds and MDPs linked to training tasks that can replicate competition demands seems necessary, with the aim of optimizing the training process to improve performance and prevent injuries.
Lines 304-306: as before it is difficult to follow the paragraph, please rephrase it.
Authors: Thank you very much for your contribution. Paragraph rewritten: For this, it is necessary to detected which are the periods of maximum demand of the matches and which of them are generated through the training tasks to replicate the demands of the competition taking the athlete to situations at the limit of his capacity.
Lines 307-310: this seems to fit more in a “Practical application” section than in the “Conclusion”, you could offer some more example of the use of your results and provide a separate paragraph of practical applications.
Authors: Thank you very much for your contribution. More example added: Moreover, other contextual factors as the result of the match, playing position and moment of the season can influence in the MDPs.
Practical application added: The main practical application of knowing these demands, its relation to apply and practice training tasks that meet these needs both individually and positionally on a day-to-day basis, not only to improve performance but also to have a protective effect on the relationship that the sprint has proven to have both.
Minor
Line 100: Wide midfielders, wide should not begin with a capital letter.
Authors: Thank you for catching the error. Corrected in the manuscript
Line 264: I guess the full point after [25,30]. is a mistake. Otherwise, the sentence would make no sense.
Authors: Thank you for catching the error. Corrected in the manuscript
Line 288: I imagine that “updated if both in training”, should be “updated it both in training”.
Authors: Thank you for catching the error. Corrected in the manuscript
Along the text and in the tables, please be consistent with the unit of measurement style. If you use the (m.min-1) than also m/s should be m.s-1. If the latter is chosen, please write it in the proper form: m·min-1 and m·s-1.
Authors: Thank you for catching the error. Corrected in the manuscript
Reviewer 2 Report
Results showed that wide midfielders covered the greatest distance >80% of the maximum speed (2.4 seg ± 1.63) and the longest duration (21.91 meters ± 13.35) in their MDP. When the team was losing showed greater distances (20.23 meters ± 13.04) and longer durations (2.24 seg ± 1.58) compared to games in which it was winning, and when the team ended up drawing, the relative sprint distance covered in the second half was significantly greater than in the first (16.12 ± 21.02; SD= 0.26 ± 0.28 (-0.03/-0.54). Current results highlighted that different demands of MDP are needed, according to the sprint variable relative to the maximum individual capacity in competition are required, when contextual game factors are considered.
Some references are of an older date, which would be good to take into the consideration.
Author Response
Results showed that wide midfielders covered the greatest distance >80% of the maximum speed (2.4 seg ± 1.63) and the longest duration (21.91 meters ± 13.35) in their MDP. When the team was losing showed greater distances (20.23 meters ± 13.04) and longer durations (2.24 seg ± 1.58) compared to games in which it was winning, and when the team ended up drawing, the relative sprint distance covered in the second half was significantly greater than in the first (16.12 ± 21.02; SD= 0.26 ± 0.28 (-0.03/-0.54). Current results highlighted that different demands of MDP are needed, according to the sprint variable relative to the maximum individual capacity in competition are required, when contextual game factors are considered.
Some references are of an older date, which would be good to take into the consideration.
Authors: Thank you very much for your contribution.

Reviewer 3 Report
Dear Authors,
I hope you are doing very well.
Congratulations for the work developed so far. I can glaze potential on this paper, but currently much more work is needed until it get suitable for publication. I hope that the comments below help you on improving your work.
Kind regards,
1 - General comment: Across the paper, please write shoter sentences and clarify the message that you intend to share. Many times the sentences are too long and thereby hard to follow. In other cases, the sentence/message is too vague.
2 - lines 14-16 - The independent variables analysed in the paper must be integrate in the purpose, like: winning or losing situations, players positions, etc...; This misalignment also occured at the introduction, but at the begining of discussion the purpose is well-written. Please, clarify the message over the paper.
3 - lines 37-42 - This sentence is too long and, thereby hard to follow. Please, revise it accordingly
4 - lines 54-57 - You started the introduction mentioning the harmsting injuries as a reason that support the pertinence of your investigation. However, your purpose, data collection or data analysis did not included any information about harmstrings.
5 - lines 85-87 - These are the INDEPENDENT variables considered.
6 - lines 129-130 - why did you use standardised mean differences with magnitude based inferences and not the common t-test or ANOVA?
7 - Data analysis - How was calculated the Cohen's d? Also, the qualitative analysis of changes have to be reported in data analysis. I mean, the ranges of classification...
8 - lines 178-179 - This information should be mentioned only at the end of this section
9 - Discussion - These first two/three paragraph are only dedicated to explain why you used MDP - something that should have done at the introduction.
10 - lines 211-231 - This information should be mentioned in the methodology, not here.
11 - line 232 - Only here you started the debate of your outcomes. Such debate must be much more extensive, deeper, and contrasting the findings of previous study everytime that you present a main finding of your study.
12 - lines 285-285 - "they found", who???
13 - lines 287-292 - again, this sentence is too long and dense.
14 - lines 303-310 - this entire section is too vague. How coaches can do that? Based on what guidelines? Please, add practical messages that could be useful for coaches in a daily basis analysis.
Author Response
Dear Authors,
I hope you are doing very well.
Congratulations for the work developed so far. I can glaze potential on this paper, but currently much more work is needed until it get suitable for publication. I hope that the comments below help you on improving your work.
Kind regards,
1 - General comment: Across the paper, please write shoter sentences and clarify the message that you intend to share. Many times the sentences are too long and thereby hard to follow. In other cases, the sentence/message is too vague.
Authors: Thank you very much for your contribution.
2 - lines 14-16 - The independent variables analysed in the paper must be integrate in the purpose, like: winning or losing situations, players positions, etc...; This misalignment also occured at the introduction, but at the begining of discussion the purpose is well-written. Please, clarify the message over the paper.
Authors: Thank you very much for your contribution. Purpose modified: The objective was to analyse the most demanding passages (MDP) considering the sprint variable relative to the maximum level of sprint ability of each player in function of player position, final outcome and part of the match, during the competitive phase of a professional soccer season
3 - lines 37-42 - This sentence is too long and, thereby hard to follow. Please, revise it accordingly
Authors: Thank you very much for your contribution. Paragraph rephrased: In recent years, high-intensity actions, as well as the distance and number of sprints, have increased by around 30-35% in official competition matches [8] so players will have to be robust enough to cope with such demands [9] and execute their tactical role effectively [10].
Morever to augmented in the intensity of the game, the rate and burden of hamstring injuries has also increased significantly since 2001, showing an average annual increase of 2.3% [11]
4 - lines 54-57 - You started the introduction mentioning the hamstring injuries as a reason that support the pertinence of your investigation. However, your purpose, data collection or data analysis did not included any information about hamstrings.
Authors: Thank you very much for your contribution. Information included.
5 - lines 85-87 - These are the INDEPENDENT variables considered.
Authors: Thank you very much for your contribution. Added information.
6 - lines 129-130 - why did you use standardised mean differences with magnitude based inferences and not the common t-test or ANOVA?
Authors: Thank you very much for your contribution. Cohen's d was used as a measure of effect size to compare standardized mean differences. This calculation tells us how many standard deviations of difference there are between the results of the two groups being compared.
7 - Data analysis - How was calculated the Cohen's d? Also, the qualitative analysis of changes have to be reported in data analysis. I mean, the ranges of classification...
Authors: Thank you very much for your contribution. The Difference standardized (Cohen) is shown in the results tables in the corresponding column between parentheses.
8 - lines 178-179 - This information should be mentioned only at the end of this section
Authors: Thank you very much for your contribution. Information mentioned at the end of discussion
9 - Discussion - These first two/three paragraph are only dedicated to explain why you used MDP - something that should have done at the introduction.
Authors: Thank you very much for your contribution. Modified this information to introduction
10 - lines 211-231 - This information should be mentioned in the methodology, not here.
Authors: Thank you very much for your contribution. Mentioned in the methodology
11 - line 232 - Only here you started the debate of your outcomes. Such debate must be much more extensive, deeper, and contrasting the findings of previous study everytime that you present a main finding of your study.
Authors: Thank you very much for your contribution. Modified this part.
12 - lines 285-285 - "they found", who???
Authors: Thank you very much for your contribution. Paragraph deleted.
13 - lines 287-292 - again, this sentence is too long and dense.
Authors: Thank you very much for your contribution. Modified this paragraph: The present study takes as reference the maximum speed, obtained through a linear test and updated it both in training and in the competition, if this value is exceeded. However, in some cases as players with a certain time of inactivity or low participation and poor competitive rhythm, this value of maximum speed obtained at a specific moment, would be higher than the real fitness that these players have for the speed. Because of, can place a relative range of 80% maximum sprint for those players that top speed are overestimate due to his low fitness, to determinate more accurate his MDPs related to maximum speed in this particular moment and guarantees greater precision in the determination of sprint workloads and associated injury risk.
14 - lines 303-310 - this entire section is too vague. How coaches can do that? Based on what guidelines? Please, add practical messages that could be useful for coaches in a daily basis analysis.
Authors: Thank you very much for your contribution. Modified this section.
Reviewer 4 Report
Firstly, I would like to recognise the authors for the data they collected. This is a very interesting topic and the results are important to publish.
Abstract
The abstract contains all the necessary information. However, it is stated that the data of 30 players was analysed while in the main text the authors stated that 22 players participated in the study. Also, in the methodology part it is stated that data was collected from 30 players, but 22 players participated. Why did you exclude the 8 players?
Lines 20-24: ‘When the team was losing showed greater distances (20.23 meters ± 13.04) and longer durations (2.24 seg ± 1.58) compared to games in which it was winning, and when the team ended up drawing, the relative sprint distance covered in the second half was significantly greater than in the first (16.12 ± 21.02; SD= 0.26 ± 0.28 (- 23 0.03/-0.54)’.
Are you referring to the whole team? The way the above sentence is written is rather confusing. Also, values on Table 3 do not match what you reported in the abstract. Please check your values and re-write the sentence in order to be clear if you are referring to the team data or positional data etc.
Introduction
The introduction is clear and leads to the rationale of the study.
Methods
The methodology is presented in detail so that someone can replicate the study if needed.
Did you include the extra time in the analysis or not?
What is the n that is used in the tables?
Table 6 is difficult to follow. You may want to improve it.
Discussion
The first 6 paragraphs of your discussion are rather descriptive without any significant information presented. I was expecting to see the most relevant findings presented in the first paragraph. You try to support why there was a need to perform the study, but this was already supported in the introduction. I would shorten the discussion part, and concentrate on the findings of your study while providing a comparison with previously published data.
Author Response
Firstly, I would like to recognise the authors for the data they collected. This is a very interesting topic and the results are important to publish.
Abstract
The abstract contains all the necessary information. However, it is stated that the data of 30 players was analysed while in the main text the authors stated that 22 players participated in the study. Also, in the methodology part it is stated that data was collected from 30 players, but 22 players participated. Why did you exclude the 8 players?
Authors: Thank you for catching the error. Corrected in the manuscript. Only 22 players participated. The information of 30 players of the abstract and the methodology was wrong.
Lines 20-24: ‘When the team was losing showed greater distances (20.23 meters ± 13.04) and longer durations (2.24 seg ± 1.58) compared to games in which it was winning, and when the team ended up drawing, the relative sprint distance covered in the second half was significantly greater than in the first (16.12 ± 21.02; SD= 0.26 ± 0.28 (- 23 0.03/-0.54)’.
Are you referring to the whole team? The way the above sentence is written is rather confusing. Also, values on Table 3 do not match what you reported in the abstract. Please check your values and re-write the sentence in order to be clear if you are referring to the team data or positional data etc.
Authors: Thank you very much for your contribution. Information specifies to the manuscript. The results refer to whole team.
Introduction
The introduction is clear and leads to the rationale of the study.
Methods
The methodology is presented in detail so that someone can replicate the study if needed.
Did you include the extra time in the analysis or not?
Authors: Thank you very much for your contribution. The extra time was analyzed in each of the matches, as data was recorded until the end of each match. Added to the manuscript.
What is the n that is used in the tables?
Authors: Thank you very much for your contribution. All results tables represent the n in numerical data in parentheses.
Table 6 is difficult to follow. You may want to improve it.
Authors: Thank you very much for your contribution. Improved Table 6 in the manuscript
Discussion
The first 6 paragraphs of your discussion are rather descriptive without any significant information presented. I was expecting to see the most relevant findings presented in the first paragraph. You try to support why there was a need to perform the study, but this was already supported in the introduction. I would shorten the discussion part, and concentrate on the findings of your study while providing a comparison with previously published data.
Authors: Thank you very much for your contribution.
Round 2
Reviewer 3 Report
Dear Authors,
Thank you for your effort in revising this paper. I honestly appreciate it. Unfortunately, the main issues pointed in the previous round of analysis remain. For that reason, I will recommend the rejection of this work.
Kind regards,
Author Response
Dear Reviewer,
Thank you for your effort in reviewing our research. We appreciate your comments for improvement. The changes suggested by you have been made. Likewise, in agreement with other reviewers, changes were made to the original manuscript. Sometimes, it is very difficult to give an appropriate response to each reviewer.
Thank you very much for your time.
Best regards,
Reviewer 4 Report
I am happy with the revisions and have not further comments or recommendations.
Author Response
Dear Reviewer,
Thank you for your effort in reviewing our research. We appreciate your comments. We look forward to the prompt publication of the manuscript.
Thank you very much for your time.
Best regards